# Morbidity and Prognostic Factors Associated with Wild Hedgehogs Admitted to a Wildlife Rehabilitation Center in Catalonia (NE Spain) from 1995 to 2020

**DOI:** 10.3390/ani14040556

**Published:** 2024-02-07

**Authors:** Rafael A. Molina-Lopez, Elena Obón, Laila Darwich

**Affiliations:** 1Wildlife Rehabilitation Center of Torreferrussa, Catalan Wildlife Service-Forestal Catalana S.A., 08130 Santa Perpètua de Mogoda, Spain; elena.obon@gencat.cat; 2Department Sanitat i Anatomia Animals, Veterinary School, Universitat Autonoma de Barcelona, 08193 Cerdanyola del Valles, Spain

**Keywords:** wild hedgehogs, prognostic factors, wildlife rehabilitation center, *Erinaceus europaeus*, *Atelerix algirus*

## Abstract

**Simple Summary:**

Wildlife rehabilitation centers are essential hubs for hedgehog conservation. In the present study, we describe the causes of mortality and morbidity of two species of wild hedgehogs, the West European hedgehog (*Erinaceus europaeus*) and the North African hedgehog (*Atelerix algirus*), over 26 years in Catalonia, Spain. There has been an absolute increase in admissions to the WRC, especially in the categories of orphaned/young and misplacement. Animals that presented with neurological and general clinical signs, such as low body condition, presented a higher risk of unassisted mortality. In addition, macroscopic respiratory and digestive lesions were observed in 54.9% and 43.9% of the necropsied animals, respectively, and were related to the death of the animal. Morbidity and prognostic factors in wildlife rehabilitation are essential for providing effective care, making informed decisions, optimizing resources, and improving rehabilitation success rates.

**Abstract:**

Wildlife rehabilitation centers (WRC) play a crucial role in the collection of data and the monitoring of hedgehog populations. The main objective of this study was to identify the morbidity and prognostic factors associated with the mortality of wild hedgehogs admitted at a WRC in Catalonia. A total number of 3397 hedgehogs admitted from 1995 to 2020 were studied. The principal cause of admission was orphaned/young category (41%) followed by misplacement (19%), natural disease (17%), and trauma (14%). The best outcomes for release were for misplacement (93.6%), orphaned/young (72.3%), and other causes (77.6%), and the lowest proportion of released animals were found for natural disease (41.4%) and trauma (44.7%) categories. The most common macroscopic findings were the respiratory and digestive lesions. Internal parasites were also prevalent in 61% of the animals but with no association with a higher mortality. In the multivariate analyses, the prognostic indicators related with the mortality outcome were the presence of systemic (OR = 3.6, CI 95%: 2.8–4.6) and neurological (OR = 4.3, CI 95%: 2.9–6.4) signs. Morbidity and prognostic factors in wildlife rehabilitation are essential for providing effective care, making informed decisions, optimizing resources, and improving rehabilitation success rates.

## 1. Introduction

Wildlife rehabilitation centers (WRC) are essential hubs for wild animal conservation, providing care, conducting research, raising awareness, and actively contributing to the preservation of animal populations in the wild. Wildlife rehabilitation data have been widely used for studies in the fields of conservation biology, animal health science, and animal welfare. The analysis of the causes of admission have allowed the detection of threats for different native species both at the individual level and at the level of broader taxa, exploring threats of wildlife, the evolution of temporal trends, and the study of the direct and indirect effects of anthropogenic factors [1,2,3,4]. On the other hand, the investigation of infectious and parasitic diseases derived from WRCs has been very useful in disease surveillance, especially in cases of zoonoses, monitoring wildlife to human or domestic animal spillover, and as a tool in risk assessment when planning the releases and reintroduction of wild animals [5,6,7,8,9,10,11]. Finally, it should be noted that wildlife rehabilitation is based on the treatment and management in captivity of free-living animals that are sick, injured and, in any case, subject to multiple stress factors. Therefore, the study of wild animal rehabilitation protocols, the analysis of the outcomes, and the post-release follow-up are essential for improving the welfare of the individuals admitted to WRCs [12,13,14,15].

Hedgehogs play a crucial role in ecosystems by controlling invertebrate populations, such as slugs, beetles, and worms, or serving as indicators of environmental health. Changes in hedgehog populations can signal problems in the environment, such as habitat degradation, pollution, or the impact of human activity. Finally, their presence contributes to biodiversity within different ecosystems, since they inhabit various habitats, from gardens to woodlands. Wildlife rehabilitation centers (WRC) often collect data and monitor hedgehog populations. These data contribute to scientific research, helping understand population trends, health status, and threats that hedgehogs face in the wild. Such information aids in the development of better conservation strategies.

Hedgehogs are the most commonly admitted mammals in wildlife rehabilitation centers (WRC) in many countries of Europe, probably because they inhabit synanthropic environments [16,17,18,19]. Two species of hedgehogs are present in Catalonia, the West European hedgehog (*Erinaceus europaeus*) and the North African hedgehog (*Atelerix algirus*) [20]. Both species are protected by the Catalan government [21] and have been classified as Least Concern (LC) by the International Union for Conservation of Nature (IUCN) Red List of Threatened Species [22,23].

In Europe, many studies support the decline of European hedgehogs in the wild [24,25]. Thus, a better understanding of the threats affecting their populations is essential for the application of mitigation measures. Anthropogenic and natural factors have an effect in individual morbidity and mortality, but also at a population level [26,27,28]. Data derived from WRCs are very useful for both the improvement in the individual welfare of animals and the conservation of the species [29,30]. 

A prognostic factor is any variable that is associated with the risk of a subsequent health outcome among individuals with a particular health condition. [31]. Prognostic studies allow the assessment of factors which relate baseline clinical variables to outcomes and need to be implemented in order to improve patient care and optimize economic resource use [32]. To carry out this type of study, it is necessary to collect data regarding demographics and clinical signs assessed at the physical exam. Whenever possible, complementary diagnostic methods (such as biopathological analysis or diagnostic imaging techniques) and the detection of parasites or other microorganisms allow a deeper investigation of prognostic factors. In veterinary medicine, prognostic factor research is growing, especially in domestic species such as dogs [33,34], cats [35], and horses [36]. However, research focused on the estimated prognostic factors of wildlife casualties are scarce [37,38].

Knowledge about hedgehog diseases has grown significantly. Various parasites have been described in hedgehogs as well as their effect on the health of parasitized animals [39,40]. Moreover, a large number of reports have been published about pathogenic agents in hedgehogs around the world, as well as their effects on the health of these species, their zoonotic potential, or their role in transmission to domestic or wild animal species [41,42,43,44,45].

The aim of this study was to analyze the causes of admission and the outcomes of wild hedgehogs admitted to a WRC in Catalonia across 26 years and to identify the prognostic factors associated with their unassisted mortality.

## 2. Materials and Methods

### 2.1. Study Design

A retrospective study was performed using the original medical records of animals admitted at the wildlife rehabilitation center (WRC) of Torreferrussa (Catalonia, northeast Iberian Peninsula). The center receives animals from the whole county of Catalonia, mainly from the north and central areas. Catalonia is an autonomous community of Spain located in the Mediterranean subregion of the western Palearctic (3°19′–0°9′ E and 42°51′–40°31 N). Wild hedgehogs admitted alive from 1995 to 2020 were included in the analyses. Captive-born hedgehogs as well as cases with incomplete data were excluded from the analysis.

The WRC is under the direction of the Catalan Wildlife Service, which stipulates animal management protocols and ethical principles according to the Catalan [46] and Spanish [47] legislations. Animals that had to be euthanized for animal welfare reasons were administrated barbiturates by intravenous injection.

The animal data section included the following variables: species, sex, and age. Sex was determined by genital inspection. Age was categorized as “First Calendar Year” or “>1 Calendar Year” according to anamnesis and morphological criteria.

### 2.2. Morbidity and Mortality Analysis

The causes of admission were based on the data collected from the admission report and the primary diagnoses [18]. Briefly, causes were subjectively grouped in the following main categories: “Trauma” (car collisions, animal attacks, and garden tools accidents), “Orphaned/young” (young animals supposedly abandoned by their parents and inexperienced juveniles), “Natural disease” (infectious or parasitic disease, starvation and weakness, and other diseases grouped by organ system), “Misplacement” (casual/accidentally found with no apparent damage), and “Other causes” (including intoxication, held in captivity, and drowning).

Clinical signs were recorded by the veterinary staff and keepers at admission and the pathological syndromes were classified by physiological organ systems: general disease (weakness, low body condition, dehydration, hypothermia, etc.), neurological (depression, ataxia, head tremor, paralysis, etc.), musculoskeletal (amputation, fractures, luxation, and lameness), integument (skin and subcutaneous conditions), and others (including behavioral abnormalities and eye and adnexa problems, diarrhea, dyspnea, etc.). We adopted a single-condition morbidity responsible for the animal’s need for therapy or investigation [48]. In addition, animals were classified as “Healthy” or “Sick”. 

Parasitological analyses: an overall inspection of stools for parasites was carried out via a direct wet preparation and a zinc sulphate flotation (specific gravity of 1.35). Eggs and larvae were identified to genus or family level by simple microscopic examination based on the hedgehog helminths that have been described in Europe [49]. Several coprological analyses were performed on each individual throughout the period of their stay in the WRC. The results of each fecal analysis were categorized with a binary variable (positive/negative) and as single or multiple infection (if more than one genus of parasite was detected).

Clinical and pathological data collected included the packed cell volume (PCV) and total solid (TS). Values of TS and PCV were estimated at the WRC by microhematocrit centrifugation and the refractometric method, respectively. Both variables were categorized as normal, low, or high, according to reference values published by Rossi et al. (2014) [50].

Necropsies were performed at the WRC by the veterinary staff. Probable cause of death was based on the macroscopic findings and was categorized by physiological organ systems. Thus, each death was recorded in terms of the organ or system determined to be primarily responsible for the death [51].

The final outcomes were categorized into four categories as follows: released, unassisted mortality, euthanized, and kept in captivity.

### 2.3. Statistical Analysis

Statistical analyses were performed with SPSS Advanced Models™ 15.0 (SPSS Inc. 233 South Wacker Drive, 11th Floor Chicago, IL 60606-6412, USA). A *p*-value of <0.05 was selected for statistical significance. Descriptive statistics were given with 95% confidence intervals (CI 95%). CI were calculated using the Macro “Confidence Interval for Proportions !CIP V2003.07.15 (c) A.Bonillo, JM.Domenech & R.Granero”. Median, 25th percentile (P_25_), and 75th percentile (P_75_) were used for describing time variables. For statistical inference, a Chi-square (χ^2^) or a Fisher exact test was used for comparison between proportions. One-way analysis of variance or the non-parametric Mann–Whitney U test, depending on the distribution of each variable (whether it is normal or not), was used to compare means. A linear regression model was applied to analyze the trend of the causes of admission through the study period. A logistic regression model (LRM) was carried out to determine which predictor variables were associated with unassisted mortality. Briefly, the outcome was categorized as “Release = 0” and “Unassisted mortality = 1”. Bivariate analysis was performed to determine which of the variables were associated with unassisted mortality. Odds ratio (CI 95%) given by the LRM was used as a measure of association between the outcome and the predictor variables. Predictor variables included in the LRM were codified as binary variables: sex, age, presence of fecal parasites, and clinical signs, which were grouped as general signs, integument, neurological, and musculoskeletal. Furthermore, two categorical variables were included: body condition (obesity, normal, thin, emaciation) and PCV (low, normal, high). The final adjusted model was fitted after the application of a forward stepwise method following the likelihood ratio (LR) criterion.

## 3. Results

A total of 3397 animals were reviewed. Of those, 3250 (95.7%) were European hedgehogs and 147 (4.3%) were North African hedgehogs. The number of admissions suffered a significant increase over the period of the study, principally in the orphaned/young (R^2^ = 0.78), natural disease (R^2^ = 0.82), and misplacement (R^2^ = 0.85) categories (Figure 1). The demographic characteristics are summarized in Table 1. 

Most of the admissions were concentrated in spring and summer (69%) and only 10% of them occurred in winter. An overall increase in young animal admissions over the period of study was observed with a second peak in autumn and winter.

### 3.1. Causes of Admission and Outcomes 

The principal cause of admission was “Orphaned/young” (41.0.0%) followed by “Misplacement” (19%), “Natural disease” (17.1%), and “Trauma” (14.0%). In the North African hedgehog, the frequencies of “Orphaned/young” (50.3%) and “Other causes” (15.6%) were statistically significantly higher (χ^2^ = 21.5; df = 4; *p* < 0.001). On the other hand, “Trauma” and “Natural disease” were more frequent in males in both species (χ^2^ = 11.5; df = 4; *p* = 0.021). The prevalences of the causes of admissions are presented in Table 2.

The lower numbers of released animals were observed for natural disease (41.4%) and trauma (44.7%) categories (Figure 2). In addition, the highest prevalence of unassisted mortality was observed in the natural disease category, representing more than 53%, and the highest rates of euthanized animals occurred in the trauma category (around 18%). In contrast, misplacement (93.6%), other causes (77.6%), and orphaned/young (72.3%) presented better outcomes with a larger proportion of released animals. 

### 3.2. Risk Factors Associated to Clinical Signs

Based on the medical records, 1394 (44%) hedgehogs were classified as sick or injured at admission. The frequencies of clinical signs grouped by organ systems were as follows: 94.4% general signs, 57.8% integument, 19.4% neurological, 10% musculoskeletal, and 12.2% other clinical signs. Table 3 shows the frequency of the clinical signs in the outcome categories and the odds ratio (CI 95%). The clinical conditions with worse prognoses were shock, paralysis, depression, and emaciation.

### 3.3. Parasitological Examination

A total number of 1187 coprological analyses were evaluated: 631 individuals had one single fecal exam, 319 had two, 159 had three, 77 had four, and 29 had five coprological tests. A proportion of 61% of the animals tested were positive in at least one analysis and 41.6%. The prevalences of infection are summarized in Table 4.

Multiple infection was observed in 26.3% of the animals, representing 49% of the infected hedgehogs. *Capillarid* nematodes (42.3%) and the lung worm *Crenosoma striatum* (41.6%) were the most common parasites. Moreover, the presence of both nematodes was the most frequent multiple infection (21.2%).

Parasite infection was not associated with sex or the species in any of the variables studied. The prevalence of infection in young hedgehogs was statistically significantly lower than adults in the following variables: overall infection (49.2% vs. 73.3%; χ^2^ = 63.1; df = 1; *p* < 0.001), *Crenosoma striatum* (30.4% vs. 37.7%; χ^2^ = 6.2; df = 1; *p* < 0.001), *Capillarid* (25.6% vs. 48.3%; χ^2^ = 57.2; df = 1; *p* < 0.001), and multiple infection (24.4% vs. 35.9%; χ^2^ = 63.2; df = 1; *p* < 0.001). Interestingly, 53.6% of the hedgehogs classified as healthy were parasitized, and 66% of the sick animals were parasitized.

### 3.4. Pathological Findings

The total number of necropsies was 989 out of 1238 corpses (80%). The median time of the necropsy was 1 day and the P_75_ was 3 days. According to the experience of the veterinarians, 66.8% of the necropsies were fresh, 29.3% had macroscopic evidence of autolysis, and 4.9% were rotten. Moreover, the worse preserved samples came from the digestive tract and the nervous system. Respiratory (54.9%) and digestive (43.9%) lesions were the most frequent. Pneumonia and verminous bronco-pneumonia together accounted for 60.5% of the respiratory conditions. Within the digestive system, gastric ulcers or erosions represented 52% of digestive pathological findings (Table 5).

When lesions were grouped by physiological organ systems, no statistically significant differences were observed between species or sexes. On the other hand, adults had higher frequency of respiratory (56.8% vs. 46.8%; χ^2^ = 8.6; df = 1; *p* = 0.004), integument (32.8% vs. 25.8%; χ^2^ = 5.3; df = 1; *p* = 0.023), and lymphatic lesions (33.4% vs. 26.9%; χ^2^ = 4.4; df = 1; *p* = 0.035) than young animals. Regardless of the cause of admission, digestive and respiratory lesions had the higher frequencies, mostly in the categories of natural disease and orphaned/young. Lesions affecting the musculoskeletal system and the integument were more frequent in trauma-related admissions. Macroscopic lesions according to the cause of admission are presented in Figure 3.

### 3.5. Prognostic Factors for Animal Survival 

A logistic regression analysis was performed to estimate the survival of the animals admitted for rehabilitation. Descriptive parameters of the prognostic factors are presented in Table 6. In the bivariate model, all variables were associated with a worse prognosis, except being female or not being parasitized. However, in the multivariate analyses, the prognostic indicators significantly related with unassisted mortality were the systemic and neurological clinical signs.

## 4. Discussion

In the present study, the results of the rehabilitation of two species of wild hedgehogs admitted to the WRC of Torreferrussa (NE Spain) over a period of 26 years are presented. First, demographic and epidemiological variables relevant to the conservation of both species are reported, such as the analysis of the causes of admission and the outcomes. However, the strong point of this work is the description of clinical variables obtained from live animals and the findings of necropsies in order to determine the prognostic factors associated with the outcomes.

A total number of 3397 hedgehogs belonging to the two native wild species admitted at a WRC were analyzed. The North African hedgehog is widely distributed in Spain, along the Mediterranean basin from the south of Catalonia to Andalusia [52]. The WRC of Torreferrussa is located on the middle of the coast of Catalonia, where the density of this species is very low. For this reason, most of the individuals admitted at the WRC were European hedgehogs. The most relevant demographic characteristics were the absence of differences between sexes in both species and the highest proportion of admissions of animals classified as “First Calendar Year”. It was not possible to use invasive techniques to determine the age of our patients, so it must be assumed that some misclassification biases occurred, because a weight overlap between adults and subadults exists [53].

Spring and summer were the seasons with the highest number of hedgehog admissions at the WRC, like previously reported in other regions of the Iberian Peninsula [54,55]. An absolute rise in the wild hedgehog population throughout the breeding season and an increase in the patterns of activity of animals could explain these results. Moreover, we also observed an increase in the second peak of hoglets because of the second litter in autumn over the years of the study [19,56].

One of the most interesting results is related to the close interaction between humans and wild hedgehogs inhabiting a territory with a high density of urban areas. Over the period of the study, we observed a rise in the number of annual admissions. A plausible explanation of these results includes the following factors: an absolute increase in casualties or animal–human interactions combined with an increase in people’s awareness of animal welfare and wildlife rehabilitation [56,57]. In the last decade, there has been a growing awareness and empathy toward wildlife due to various reasons, including environmental education, media coverage, and an understanding of the impact of human activities on ecosystems. Thus, people are more likely to rescue injured, orphaned, or distressed animals and to bring them to the WRC.

A clear definition of the causes of morbidity is crucial to be able to compare results between WRCs. For example, in the present study, the term “Misplacement” is described as “casual/accidentally found with no apparent damage”. Other authors have used different names referring to this cause such as “casual encounter” [54], “random finds” [55], or “intrusion or unnecessary capture” [57]. Although the meanings of the variables in those works are clearly described in the methodology, the adoption of an agreed nomenclature will be a very useful tool for future research. In this context, a more detailed analysis of the circumstances of the collection of each hedgehog is necessary for assigning a less ambiguous cause of admission, whenever possible. In fact, the International Classification of Diseases (ICD) offers a model that could be adapted to veterinary medicine [58].

It is well documented that anthropogenic factors encompass the most prevalent causes of admission at WRCs worldwide, representing up to 31% [4,59] to 39.8% [2]. Direct anthropogenic causes, such as accidents or trapping, must be differentiated from indirect causes or interactions, such as the accidental discovery of adult or young specimens in human settlements or the discovery of healthy puppies or those in a vulnerable situation due to the destruction of nests or the disappearance of parents. On the other hand, non-anthropogenic causes of admissions such as weakness or malnourishment and infectious or parasitic disease are factors of major concern for the success of the rehabilitation process. 

Our study is in agreement with the work of Lukesova [57], with “Orphaned/young” being the most frequent cause of admission. As in other wildlife species, a proportion of young hedgehogs are removed from the wild and brought to WRC unnecessarily due to public ignorance [60]. However, it is crucial to identify the hoglets that need specialized human care to ensure their survival. Signs indicating that the animal needs help are the finding of isolated juveniles weighing less than 200 g, being active during the day, vocalizing, or showing symptoms of illness or trauma [61].

The crude analysis of the outcomes of hedgehog rehabilitation between centers shows that more than 40% of hedgehogs are released [62,63] and that unassisted mortality is around 30% in studies conducted in Portugal (33%) [55], the Czech Republic (25%) [57], or Spain (28%) [54]. If we intend to compare these results in terms of effectiveness, it is necessary to carry out a stratified analysis according to the causes, the clinical diagnoses, or the prognostic category [48]. 

An approach to the outcome analyses based on the study of clinical signs will allow the identification of prognostic factors, helping to improve veterinary care and animal welfare [64]. Regardless of the cause of admission, the highest prevalence of clinical signs in sick or injured hedgehogs corresponded to general clinical signs such as low body condition, hypothermia, and weakness. All of them are associated with a higher risk of natural death. Therefore, it is necessary that wildlife rehabilitators be trained to carry out a complete physical examination and collect clinical variables such as weight and body condition, temperature, or degree of hydration and to recognize the most common symptoms in hedgehogs.

In our cohort, 61% of hedgehogs were positive to internal parasites, with 26% of infections caused by more than one type of parasite. A large variety of internal parasites has been described in hedgehogs, probably explained by their omnivorous diet. They ingest many invertebrates that can act as paratenic or intermediate hosts of various parasites [40]. 

The overall prevalence of parasites was as high as those reported in Poland (74%) [65] and in the UK (69%) [49]. *Capillarid* and *Crenosoma striatum* larvae were the most common nematodes, with an overall prevalence of 42.3% and 41.6%, respectively. These parasites were also the most frequent in coprological analyses carried out in Germany over a long period of time [66,67,68] and in samples from hedgehogs admitted to WRCs in Greece [69]. Sex was not associated with the risk of infection as described in European hedgehogs in Spain [70] and Britain [49], but the prevalence was greater in adults, which is probably explained by an age-dependent parasite accumulation [49]. Infected hedgehogs may present with a range of clinical signs such as weight loss, nasal discharge, dyspnea, wheezing, cough, and exercise intolerance [71,72]. 

Mortality is an indicator of hospital performance quality in human health care [73]. Moreover, necropsy is a tool for determining the cause of death to discover underlining pathologies, and it is essential for disease surveillance [74,75]. In our WRC, necropsies are routinely performed by the veterinary staff both in cases of unassisted mortality and in euthanized animals. Most hedgehogs (80%) were necropsied as soon as possible, because decomposition artifacts are commonly observed in this species. The highest rate of mortality was found for natural disease (59%) and trauma (55%). Similar results have been reported in France, where trauma (41%) and bacterial infections (34%) were also the principal causes of death [76], or in Portugal, which also reported trauma (33%) as the main cause [55]. Unfortunately, this study is based on macroscopic findings, and future research reviewing histopathological reports will allow a deeper discussion.

On the other hand, the causes of death attributable to respiratory diseases (32.7%) were the most frequent followed by nonspecific (28.9%) and digestive (15.1%). Verminous pneumonia may frequently result in fatal disease for hedgehogs, and it is one of the most common pathologies in hedgehogs [77]. Regarding digestive pathology, gastric ulcers or erosions accounted for 52% of the lesions, although the cause remains undetermined. In other species, stress, bacterial or parasitic infection, and intoxication could be involved [78,79].

In fact, digestive and respiratory pathological changes were the most frequent in any of the causal categories. This finding is highly suggestive of underlying diseases not diagnosed at admission or the acquisition of illness during the stay at the center, possible secondary to stress, nutritional deficiencies, or contact with conspecifics [80]. 

Finally, the worst prognostic factor was related with neurological conditions (OR = 7.26) including paralysis (OR = 12.77) and depression (OR = 7.40). Similar results have been described in birds of prey [81] and in eastern gray squirrels [82]. Emaciation (OR = 7.20), dyspnea (OR = 6.21), and hypothermia (OR = 5.17) also are related with higher mortality. Many factors are associated with survival to the release of the hedgehogs and the success of the therapy, such as extent and severity of injuries [37], individual stress, or parasitic or bacterial infections which worsen the prognosis [72]. Systemic signs of diseases, including low body condition and neurological signs, were the clinical presentation related with higher mortalities, with little change in the odds ratio in the final multivariate model. Thus, triage and treatment protocols must be based on a good physical examination. To accomplish this, the appropriate training of the staff is essential [83]. 

European hedgehogs are the most common mammals admitted to WRCs in Europe, due to their affinity with humans and the ease with which they can be captured. A reduction in population has been reported in Europe. WRCs allow not only the recovery of individuals and the reinforcement of populations locally, but also provide data that allow a better understanding of the threats to this species and their role as sentinels under the One Health approach [84]. Research into morbidity and prognostic factors contributes to the development and refinement of standardized protocols. It helps establish evidence-based best practices that can be shared among different rehabilitation centers to promote uniformity and effectiveness in care. 

## 5. Conclusions

Throughout the 26 years of the study, there has been an absolute increase in admissions to the WRC, especially in the categories of orphaned/young and misplacement. This fact is explained by the cohabitation of these species with humans and the ease of being captured by people. However, 44% of hospitalized hedgehogs were classified as sick and presented a higher risk of unassisted mortality, especially in the case of neurological alterations and general signs, such hypothermia or low body condition. In addition, macroscopic respiratory and digestive lesions were observed in 54.9% and 43.9% of the necropsied animals, respectively, and were related to the death of the animal. The study of morbidity and prognostic indicators in the rehabilitation of wild hedgehogs plays a pivotal role in identifying risk factors that contribute to illness, injury, or unsuccessful rehabilitation in hedgehogs, refining and standardizing protocols, improving outcomes, and ultimately contributing to the conservation and welfare of these wild animals.

## Figures and Tables

**Figure 1 animals-14-00556-f001:**
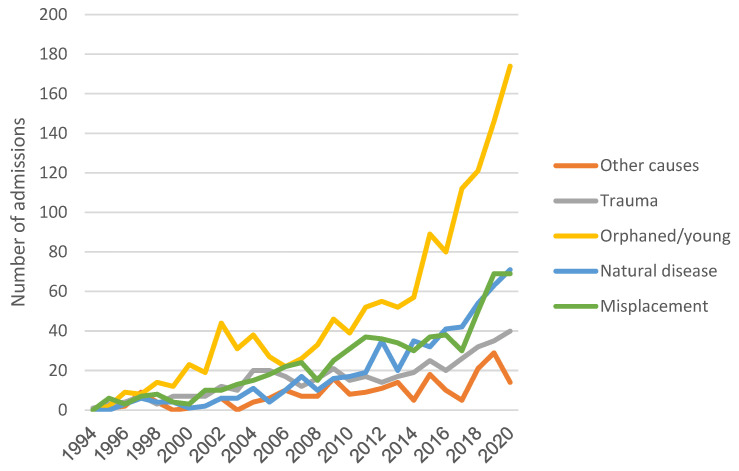
Annual admissions throughout the period of study.

**Figure 2 animals-14-00556-f002:**
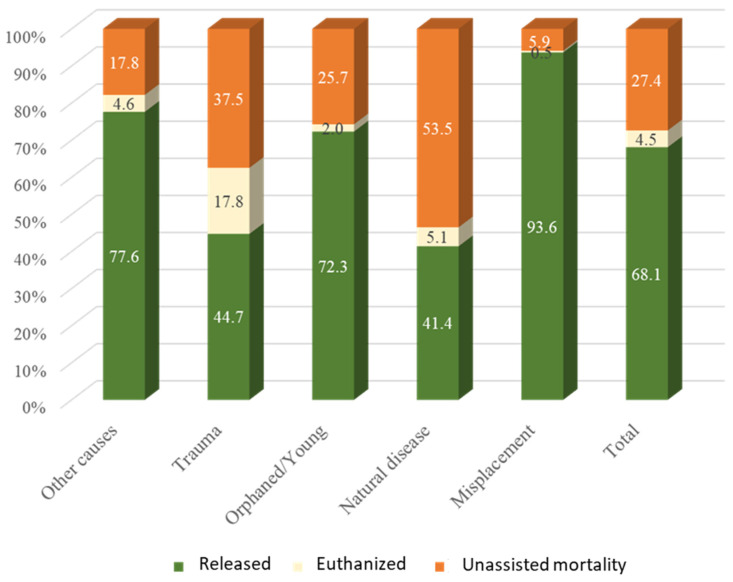
Frequency of outcomes relative to the cause of admission.

**Figure 3 animals-14-00556-f003:**
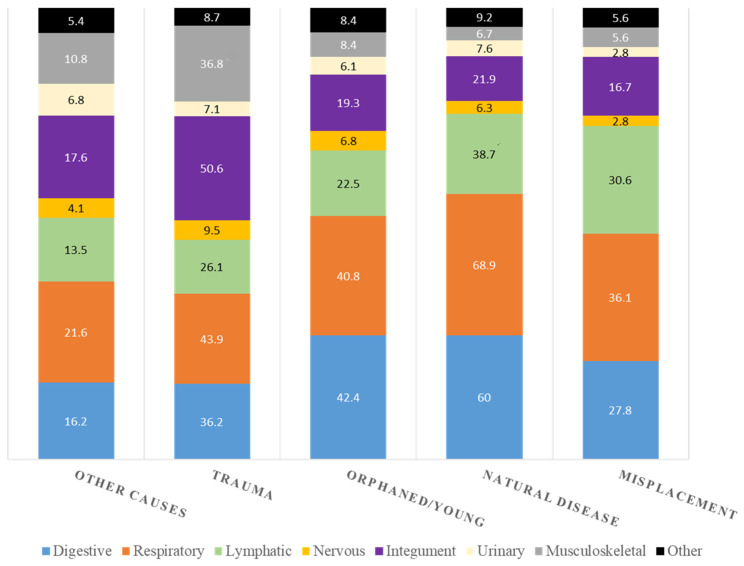
Macroscopic lesions grouped by organ systems according to the cause of admission.

**Table 1 animals-14-00556-t001:** Demographic characteristics of the individuals included in the study.

Species	Total	Sex (*n,* %)			Age (*n*, %)		
	*N*	Unknown	Male	Females	Unknown	>1 Calendar Year	First Calendar Year
*Erinaceus europaeus*	3250	428 (13.2)	1441 (44.3)	1381 (42.5)	91 (2.8)	1207 (37.1)	1952 (60.1)
*Atelerix algirus*	147	20 (13.6)	64 (43.5)	63 (42.9)	6 (4.1)	49 (33.3)	92 (62.6)
Total	3397	448 (13.2)	1505 (44.3)	1444 (42.5)	97 (2.9)	1256 (37.0)	2044 (60.2)

**Table 2 animals-14-00556-t002:** Prevalences of the causes of admission. The information showed in the cells represents the number of animals, the prevalence, and the confidence intervals (95%).

Cause of Admission	Others (*n*; P (CI 95%)	Trauma (*n*; P (CI 95%)	Orphaned/Young (*n*; P (CI 95%)	Natural Disease (*n*; P (CI 95%)	Misplacement (*n*; P (CI 95%)
Overall	303; 8.9 (7.9–9.9)	474; 14.0 (12.8–15.1)	1393; 41.0 (39.3–42.6)	582; 17.1 (15.9–18.5)	645; 19.0 (17.7–20.3)
*Erinaceus europaeus*	280; 8.6 (7.7–9.6) *	456; 14.0 (12.8–15.3)	1319; 40.6 (38.9–42.3) *	571; 17.6 (16.3–18.9)	624; 19.2 (17.8–20.6)
*Atelerix algirus*	23; 15.6 (10.2–22.5) *	18; 12.2 (7.4–18.7)	74; 50.3 (41.0–58.7) *	11; 7.5 (3.8–13)	21; 14.3 (9.1–21.0)
Male	95; 48.0 (40.8–55.2)	227; 53.0 (48.1–57.5) *	643; 53.3 (50.5–56.2)	281; 51.7 (47.4–55.9) *	259; 45.3 (41.1–49.5)
Female	103; 52.0 (44.8–59.2)	202; 47.1 (42.3–51.9) *	563; 46.7 (43.8–49.5)	263; 48.3 (44.1–52.6) *	313; 54.7 (50.5–58.9)

* Statistically significant differences (*p* < 0.05). Chi-square test (χ^2^). Degree of freedom (df) = 4.

**Table 3 animals-14-00556-t003:** Prognostic factors associated with unassisted mortality based on clinical signs grouped by organ system.

Clinical Sign Descriptor	Released *n* (%)	Unassisted Mortality *n* (%)	Odds Ratio (CI 95%)
General signs	735 (56.0)	578 (44)	3.9 (3.3–4.6)
Emaciation	87 (31.9)	186 (68.1)	7.2 (5.4–9.6)
Weakness	519 (54.0)	442 (46.0)	3.3 (2.8–3.9)
Hypothermia	89 (36.2)	157 (63.8)	5.2 (3.9–6.8)
Dehydration	215 (50.6)	210 (49.4)	2.9 (2.4–3.6)
Shock	7 (9.6)	66 (90.4)	25.4 (11.6–55.7)
Hemorrhage	16 (39.0)	25 (61.9)	3.9 (2.1–7.5)
Dyspnea	27 (30.0)	63 (70.0)	6.2 (3.9–9.8)
Neurological	67 (29.1)	163 (70.9)	7.3 (5.4–9.8)
Depression	62 (28.6)	155 (71.4)	7.4 (5.4–10.1)
Ataxia	5 (31.3)	11 (68.8)	5.6 (1.9–16.0)
Paralysis	4 (16.7)	20 (83.3)	12.8 (4.3–37.5)
Integument	453 (62.3)	274 (37.7)	1.7 (1.5–2.1)
External parasites	270 (62.2)	164 (37.8)	1.6 (1.3–2.0)
Skin diseases	187 (60.1)	124 (39.9)	1.8 (1.4–2.3)
Soft tissues	94 (51.6)	88 (48.4)	2.5 (1.8–3.4)
Musculoskeletal	62 (57.9)	45 (42.1)	1.8 (1.2–2.7)
Lameness	43 (59.7)	29 (40.3)	1.7 (1.1–2.7)
Fracture	30 (56.6)	23 (43.4)	1.9 (1.1–3.4)
Others	89 (58.2)	64 (41.8)	1.9 (1.3–2.6)
Oral/dental disorders	13 (52.0)	12 (48.0)	2.3 (1.1–5.1)
Diarrhea	4 (28.6)	10 (71.4)	6.3 (2.0–20.2)

**Table 4 animals-14-00556-t004:** Prevalence of fecal parasites.

Parasite	*N*	Prevalence (CI 95%)
*Crenosoma striatum*	494	41.6 (38.8–44.5)
*Capillaria* spp.	502	42.3 (39.5–45.2)
*Coccidia*	102	8.6 (7.1–10.3)
*Brachylaemus erinacei*	31	2.6 (1.8–3.7)
Others	8	0.6 (0.03–1.1)
Positive (binary)	723	60.9 (58.1–63.7)
Single infection	369	31.1 (28.5–33.8)
Multiple infection	313	26.3 (23.9–28.9)

**Table 5 animals-14-00556-t005:** Prevalence of macroscopic lesions grouped by organ systems.

Macroscopic Lesions	*n*	Prevalence (CI 95%)
**Respiratory system**	**543**	**54.9 (51.7–58.0)**
Pneumonia	216	39.7 (35.6–44.0)
Verminous broncho-pneumonia	113	20.8 (17.5–24.5)
Lung congestion	68	12.5 (9.8–15.6)
Lung hemorrhage	42	7.8 (5.6–10.3)
Autolysis	41	5.9 (4.1–8.2)
Lung pallor	32	5.9 (40.1–8.2)
Hemothorax	11	2.0 (1.2–3.6)
**Digestive system**	**435**	**43.9 (40.9–47.1)**
Gastric ulcers/erosions	227	52.2 (47.4–56.9)
Autolysis	102	23.4 (19.5–27.7)
Hemorrhagic enteritis	75	17.2 (13.8–21.1)
Oral/dental disorders	10	2.3 (1.1–4.2)
**Lymphatic system**	**357**	**36.2 (27.1–37.0)**
Pale spleen	114	31.9 (27.1–37.0)
Splenomegaly	82	22.9 (18.7–27.7)
Autolysis	67	18.7 (14.8–23.2)
Spleen small	22	6.2 (3.9–9.2)
Enlarged lymph nodes	12	3.4 (1.7–5.8)
Spleen laceration	12	3.4 (1.7–5.8)
**Integument (skin and soft tissue)**	**312**	**31.5 (28.6–34.5)**
Wound	140	44.9 (39.3–50.6)
Subcutaneous hematomas	37	11.9 (8.5–16.0)
Autolysis	36	11.5 (8.2–15.6)
**Musculoskeletal system**	**215**	**21.7 (19.2–24.4)**
Autolysis	65	30.2 (24.2–26.8)
Fracture	59	27.4 (21.6–33.9)
Muscle laceration	48	22.3 (16.9–28.5)
Muscular hemorrhage	21	9.8 (6.1–14.5)
**Nervous system**	**120**	**12.1 (10.2–14.3)**
Autolysis	51	42.5 (33.5–51.8)
Brain congestion	39	32.5 (24.2–41.6)
Brain trauma	15	12.5 (7.2–19.8)
Brain hemorrhage	10	8.3 (4.1–14.8)
**Urinary system**	**115**	**11.6 (9.7–13.8)**
Others *	145	14.6 (12.5–17.0)

* Others include internal hemorrhage, pericarditis, ocular lesions, diaphragmatic hernia.

**Table 6 animals-14-00556-t006:** Prognostic indicators associated with unassisted mortality of hedgehogs. Variables with statistical significance only are represented in the bivariate and multivariate regression model.

Variable Description	N	B	SE	Odds Ratio (CI 95%)	Odds Ratio (CI 95%)
				Bivariate model	Multivariate model
Age	1851	0.478	0.087	1.6 (1.4–1.9)	ns
Sex	1303	−0.173	0.086	0.8 (0.7–0.9)	ns
Clinical signs:					
General signs	1313	1.367 (1.283) ^§^	0.085 (0.122) ^§^	3.9 (3.3–4.6)	3.6 (2.8–4.6)
Emaciation	273	1.974	0.150	7.2 (5.4–9.7)	ns
Integument	727	0.560	0.090	1.7 (1.5–2.1)	ns
Neurological	230	1.983 (1.461) ^§^	0.152 (0.199) ^§^	7.2 (5.4–9.8)	4.3 (2.9–6.4)
Musculoskeletal	107	0.608	0.200	1.8 (1.2–2.7)	ns
Fecal parasites	452	−0.389	0.176	0.7 (0.5–0.9)	ns
Packed cell volume (L/L)	72	1.569	0.294	4.8 (2.7–8.5)	ns

ns, no significant differences (*p* > 0.05). ^§^ Multivariate regression coefficients.

## Data Availability

The data presented in this study are available on request from the corresponding author. The data are not publicly available unless requested from the public entity responsible for WRC.

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
