# Peer review of "Morbidity and Prognostic Factors Associated with Wild Hedgehogs Admitted to a Wildlife Rehabilitation Center in Catalonia (NE Spain) from 1995 to 2020"

_animals, 2024, doi:10.3390/ani14040556_

Round 1

Reviewer 1 Report

Comments and Suggestions for Authors

I wish firstly compliment with Authors for the effort in providing essential data usually not exploited but that represent an easy and efficient way to highlight threath and concern in wildlife species conservation and understand population trend. The paper provides a well and interesting overview on morbidity and prognostic factors associated with hedgehogs recovered at WRC usefull to understand health population status and provide new knowledge tool for the development of conservation strategies. 

I wish kindly invite the Authors to provide a more widely and detailed background on the importance in collecting this kind of data from WRC with example also on other species. The absence of the reference list did not allow me to verify if all relevant references have been included in the manuscript.

Comments on the Quality of English Language

A moderate editing of English language is required

Author Response

Dear reviewer,

Thank you very much for your constructive revision of the manuscript. We hope the revised version is containing the improvements required.

Reviewer 1 comments:

I wish firstly compliment with Authors for the effort in providing essential data usually not exploited but that represent an easy and efficient way to highlight threath and concern in wildlife species conservation and understand population trend. The paper provides a well and interesting overview on morbidity and prognostic factors associated with hedgehogs recovered at WRC usefull to understand health population status and provide new knowledge tool for the development of conservation strategies. 

I wish kindly invite the Authors to provide a more widely and detailed background on the importance in collecting this kind of data from WRC with example also on other species.

Answer: We have included this paragraph in the “Introduction”.

Wildlife Rehabilitation data have been widely used for studies in the fields of conservation biology, animal health science and animal welfare. The analyzes of the causes of admission have allowed the detection of threats for the different native species both at the individual level and at the level of broader taxa, exploring threats of wildlife, the evolution of temporal trends and the study of the direct and indirect effect of anthropogenic factors [1,2,3,4]. On the other hand, the investigation of infectious and parasitic diseases derived from WRCs has been very useful in disease surveillance, especially in cases of zoonoses, monitoring wildlife to human or domestic animal spillover and as a tool in risk assessment when planning the releases and reintroduction of wild animals [5,6,7,8,9,10,11]. Finally, it should be noted that wildlife rehabilitation is based on the treatment and management in captivity of free-living animals that are sick, injured and, in any case, subjected to multiple stress factors. Therefore, the study of wild animal rehabilitation protocols, the analysis of the outcomes and the post-release follow-up are essential for improving the welfare of the individuals admitted to WRC [12,13,14,15].

The absence of the reference list did not allow me to verify if all relevant references have been included in the manuscript.

The references section is already included.

Reviewer 2 Report

Comments and Suggestions for Authors

The manus is well written and methods and results clearly presented. The data material is very large and the conclusions therefore rest on a sound basis. I would just like to make two comments.

Line 17 Why do species always need to have a practical function? Can't they just have permission to be present.

line 34 What causes respiratory and digestive lesions? Is it antropogenetic causes. Pesticides and do they eat things that causes traume to the digestive chanel?

line 90. I do not understand the term: Misplacement (casual/accidentially found with no appearant damage)? Why are they then collected? Were they accidentially collected by humans and brought to rehabilitation?

And just a curious question: Do you know how long relised hedgehogs survive after release?

Author Response

Dear reviewer,

Thank you very much for your constructive revision of the manuscript. We hope the revised version is containing the improvements required.

Reviewer 3 comments:

The manus is well written and methods and results clearly presented. The data material is very large and the conclusions therefore rest on a sound basis. I would just like to make two comments.

Line 17. Why do species always need to have a practical function? Can't they just have permission to be present.

Answer: I agree that no animal needs permission to exist nor does it need to justify its existence. If we have included this sentence in the introduction it is to contextualize its function in the ecosystem.

Line 34.  What causes respiratory and digestive lesions? Is it antropogenetic causes. Pesticides and do they eat things that causes traume to the digestive chanel?

Answer: We expose briefly the causes of digestive and respiratory lesions in the discussion. In most of the cases, the causes of diseases are natural (infectious or parasitic). Pesticides or foreign objects could cause gastric irritation, but we have not check for pesticides in this paper.

Line 90. I do not understand the term: Misplacement (casual/accidentially found with no appearant damage)? Why are they then collected? Were they accidentially collected by humans and brought to rehabilitation?

Exactly. The animals are not “accidentally” collected. People collect “actively” the animal assuming that something wrong is happening with them. In most of the cases these animals are healthy “neighboors”, but an important proportion are sick and the public is not trained for recognize signs of illness.

And just a curious question: Do you know how long realised hedgehogs survive after release?

Answer: Unfortunately we don’t have budget for radiotracking the released animals. In most of the cases hedgehogs are “soft released”. That means than they kept in supervised facilities for a short period of time before the complete release to the wild.

Reviewer 3 Report

Comments and Suggestions for Authors

The paper addresses and interesting issue but I believe that at this stage it is not mature enough to be published. It requires effort to improve clarity, as the English is not good in some parts and the methods are not clearly explained. Additionally, the paper needs a true discussion (the current discussion mainly repeats results) and conclusions.

Simple Summary: It does not provide information about its own results; it only includes general sentences that could be used in an introduction, not in a summary

Introduction is too short. Paragraph in simple summary and some paragraphs in discussion would be better placed in introduction

L81-82: “fallen from their nest” is a sentence that seems t refer to birds

L108: “The final outcomes were categorized into four entities”, I believe categories would fit better here

L108: The term 'natural death' seems incongruous when the animal has entered the center due to a human-caused incident, such as trauma. Later in the text (L119) it is call unassisted mortlity.

L109: add comma after death

L117-122: writing needs to be improved, the methods used are difficult to understand.

L119: “with unassisted mortality using a logistic regression model”. Dichotomic response variable should be explained unambiguously. To which final result is 1 assigned, and to which one is 0 assigned?

L121-122: “application of the multivariate adjusted model.” This is difficult to understand, I guess the authors refer to a logistic regression with several predictor variables, but how was this model constructed? Including all variables? Or using some stepwise procedure?

L121: how was co-linearity tested?

L125-127: The number of admissions increased along the period of study, but it would be interesting to analyze if the relative importance of each cause changes (i.e. the percentage of each cause changes or remains stable).

L135-137: both species? This information should be given in a table or graphic

L143: degrees of freedom should be indicated. Use point as decimal separator

L145-146: “Moreover, males were a major risk of trauma casualties and natural disease than females” Change to: were at a major risk …

L168-173: paragraph difficult to understand, writing should be improved

L147-152: writing of the whole paragraph needs to be improved

Table 2: always use the same number of decimals in all table cells, in the current version sometimes decimals are lacking and coexist 14 and 14.0 for instance

Table 2: “* Statistically significant differences (p<0.05)” it is not clear among which groups the differences are

Table 2: In the male and female rows, are both species mixed? If that's the case, it doesn't make much sense, as there are differences between species

Table 3: calculation of odds ratios is not explained

Table 4. Apparently, both species are mixed here, and no differentiation has been made according to sex or age. However, in L187, it is mentioned that young individuals have a lower prevalence of certain parasites. The statistical test that should support this assertion is not provided, and possible differences between species are not analyzed.

L195: P75 not defined

Figure 3: “* Statistically significant differences (p<0.05)” it is not clear among which groups the differences are

L206: Data according to the period of the year are not presented anywhere

L219-224: paragraph difficult to understand, writing should be improved

Table 5: The table text is difficult to understand, its writing needs improvement. More explanations should be provided, such as the meanings of words within parentheses in categories like sex and age etc.

Table 5: “ns, no significant differences (p<0.05)” should be p>0.05

Table 5: regression coefficients are only presented for bivariate models

Discussion:

L230-241: text disconnected of results, may be eliminated

L293-306: Largely repeats results

L293-306: repeats methods and results, it is not a true discussion

L330-333: OR not defined

L342-353: introduction-like paragraph, not discussion

L355-358: these are not true conclusions derived from results presented in the manuscript, but some general statements that could be used in an introduction including some references.

Comments on the Quality of English Language

Comments on the quality of English are included with the general comments

Author Response

Dear reviewer,

Thank you very much for your constructive revision of the manuscript. We hope the revised version is containing the improvements required.

Reviewer 4 comments:

The paper addresses and interesting issue but I believe that at this stage it is not mature enough to be published. It requires effort to improve clarity, as the English is not good in some parts and the methods are not clearly explained. Additionally, the paper needs a true discussion (the current discussion mainly repeats results) and conclusions.

Simple Summary: It does not provide information about its own results; it only includes general sentences that could be used in an introduction, not in a summary.

Lines 12-22: Summary has been rewritten.

Introduction is too short. Paragraph in simple summary and some paragraphs in discussion would be better placed in introduction.

Lines 41-96:  Introduction has been rewritten. New references have been added.

L81-82: “fallen from their nest” is a sentence that seems to refer to birds:

L127-128: We have change de text: “orphaned (young animals supposedly abandoned by their parents and inexperienced “juveniles”

L98:  We have change the text: L140: In addition, animals were classified as “Healthy” and “Sick” .

L108: “The final outcomes were categorized into four entities”, I believe categories would fit better here: “Categories”. L155: We have change “entities” for “Categories”.

L108: The term 'natural death' seems incongruous when the animal has entered the center due to a human-caused incident, such as trauma. Later in the text (L119) it is call unassisted mortality. L156: We have change “Natural Death” for “Unassisted mortality” in all the manuscript.

L109: add comma after death. Done.

L117-122: writing needs to be improved, the methods used are difficult to understand.

L119: “with unassisted mortality using a logistic regression model”. Dichotomic response variable should be explained unambiguously. To which final result is 1 assigned, and to which one is 0 assigned?

L121-122: “application of the multivariate adjusted model.” This is difficult to understand, I guess the authors refer to a logistic regression with several predictor variables, but how was this model constructed? Including all variables? Or using some stepwise procedure?

L121: how was co-linearity tested?

The section 2.3 Statistical analysis has been rewritten.

L125-127: The number of admissions increased along the period of study, but it would be interesting to analyze if the relative importance of each cause changes (i.e. the percentage of each cause changes or remains stable).

L135-137: both species? This information should be given in a table or graphic.

Answer: If L135-137, refers to Figure 1, we have have presented the data for both species together, since the number of Atelerix specimens is very low in some of the categories.

L143: degrees of freedom should be indicated. Use point as decimal separator.

Answer: Lines 217-220. These results are presented in Table 2. Significant differences are indicated with an asterisk (*). Degrees of freedom (df) are 4.

L145-146: “Moreover, males were a major risk of trauma casualties and natural disease than females: Line 201:  Change done.

L168-173: paragraph difficult to understand, writing should be improved. L225-230: The paragraph has been rewritten.

L147-152: writing of the whole paragraph needs to be improved. L197-203.  The paragraph has been rewritten.

Table 2: always use the same number of decimals in all table cells, in the current version sometimes decimals are lacking and coexist 14 and 14.0 for instance.

Table Corrected.

Table 2: “* Statistically significant differences (p<0.05)” it is not clear among which groups the the differences are.

The text section has been rewritten and those results are presented clearly.

Table 2: In the male and female rows, are both species mixed? If that's the case, it doesn't make much sense, as there are differences between species.

In both species there were SS differences among sexes in the frequency of trauma and natural disease. We decide to show the frequency differences in the overall group.

Table 3: calculation of odds ratios is not explained. Answer: Odds calculation in explained the Statistical Section

Table 4. Apparently, both species are mixed here, and no differentiation has been made according to sex or age.

Answer: We have not found any statistical significant difference among species or sex. Moreover, the number of fecal analyses done to Atelerix is very low: 19 the first, 9 (second), 9 (third). In order to make Table 4 more simple we decide to show the results mixed.

However, in L187, it is mentioned that young individuals have a lower prevalence of certain parasites. The statistical test that should support this assertion is not provided, and possible differences between species are not analyzed. Answer: Evidence are presented in the test.

L195: P75 not defined. Answer: Defined in statistical section.

Figure 3: “* Statistically significant differences (p<0.05)” it is not clear among which groups the differences are. Answer: We have presented only the prevalence. Statistically significant differences are not shown.

L206:  Data according to the period of the year are not presented anywhere.

Answer: This line has been deleted:  According to the period of the year, in winter the presence of digestive lesions (53,3%) surpassed the proportion of no lesions (46,7%).

L219-224: paragraph difficult to understand, writing should be improved. L262-269: We have summarized some results in the text.

Table 6: The table text is difficult to understand, its writing needs improvement. More explanations should be provided, such as the meanings of words within parentheses in categories like sex and age etc.

Table 6: “ns, no significant differences (p<0.05)” should be p>0.05.

Table 6: regression coefficients are only presented for bivariate models.

Table 6 has been corrected.

Discussion:

L230-241: text disconnected of results, may be eliminated.

L293-306: Largely repeats results

L293-306: repeats methods and results, it is not a true discussion.

L330-333: OR not defined. Described in the statistical section.

L342-353: introduction-like paragraph, not discussion

L355-358: these are not true conclusions derived from results presented in the manuscript, but some general statements that could be used in an introduction including some references.

Discussion and Conclusion has been rewritten.

Round 2

Reviewer 3 Report

Comments and Suggestions for Authors

The paper has improved, particularly the discussion; however, I don't believe it qualifies for acceptance or minor revision yet. Many points still remain unclear and require further improvement. The description of statistical analyses has improved, but it is not entirely clear and needs additional explanations.

L85: “health outcome among people”, as the paper is about an animal species, it would be better to use the term 'individuals' instead of 'people.

L141-146. This paragraph is confusing, explain clearly that it refers to coprological analysis and specify how many times each individual was tested..

L164-165: “One-way analysis of variance and non-parametrical Mann-Whitney U test were used to compare means” explain in which situations the Mann-Whitney test was used instead of ANOVA

L167-168: “A logistic regression model (LRM) was carried out to determine which predictor 167 variables were associated with unassisted mortality” It should be clarified which variables are considered predictor variables. Based on Table 3, it may be inferred that the predictor variables include the presence of a clinical sign (as the odds ratio is presented in this table). However, it is unclear whether the predictor variables also include those listed in Table 6 or a combination of both. This ambiguity is confusing and should be explicitly stated in the description of the statistical methods.

L172: “the independent variables” They have been called previously predictor variables, keep uniformity in terms used

L173-174: What threshold of the Multiple Correlation Coefficient was used to determine the presence of collinearity?

L174: “multivariate analyses”. There is not clear to which analysis the authors refer. There are lots of types of multivariate analyses. Presumably the authors mean a Logistic regression with several predictors. This should be stated clearly.

L199-202: “had statistical 199 significant higher frequencies in Algerian hedgehog compared to European hedgehog 200 (χ2=21.5 p=0.001)”. “Moreover, males were a(t) major risk of “Trauma” and “Natural disease” 201 than females in both species (χ2=11.5; p=0.021)”. The authors base their affirmations relative to differences in specific causes of admission on the X2 test. However, these tests only inform that the distributions of causes differ between species or sexes, respectively, not on the particular causes that generate the difference. In fact, the authors have based their statements on differences in specific causes on the comparison of the confidence intervals of the percentage of each cause. Therefore, the text requires correction and improvement. Furthermore, there is a lack of explanation regarding how the confidence intervals were calculated.

L220: Mention of “Statistically significant differences (p<0.05). Degree of freedom (df)= 4.” does not make sense without specifying the type of statistical test to which it refers.

L235: When you mention 632 cases, 321, etc., are you referring to individuals? If so, make it clear that 632 individuals had a single fecal exam, and so on.

L245-247: The degrees of freedom for the X2 test should be provided. Additionally, maintain consistency in the number of decimals for p-values throughout the paper. If a p-value is very low, it may be presented as p<0.001.

Table 4. header “Prevalence (IC 95%)”. IC is in Spanish, it should be CI

Table 4: This table is somewhat confusing and requires a clearer explanation. Do 'First,' 'Second,' etc., tests refer to the first test, second test, etc.? If so, why do you mention a maximum of 4 tests in line 236, while the table indicates 5? Adding sample sizes in the header of each column would enhance clarity. Additionally, the rationale for presenting this table is not evident, as the information it provides is not used in a clear manner. Furthermore, the paragraph from lines 243-248 makes comparisons based on age, but age is not shown in Table 4.

L266-270 and table 5: Statistical analysis compare adult and young individuals but age is not included in table 5. X2 tests do not include degrees of freedom. In addition, comments on X2 test in L199-202 are also applicable here.

L271-273: no test for such sentence, only some unexplained asterisks in figure 3.

L280: “In the bivariate model, females and tested negative for parasites were protective factors.” Sentence difficult to understand, rewrite.

Table 6. L288-290: this is confusing, I assume that refers to the reference level of each factor, but this should be clarified.

Comments on the Quality of English Language

included in the general comments

Author Response

The authors would like to express their gratitude for the excellent review work that the reviewer has done on the manuscript, which has undoubtedly contributed to the improvement of the manuscript. All the comments have been introduced in the second review round of the manuscript as follows:

L85: “health outcome among people”, as the paper is about an animal species, it would be better to use the term 'individuals' instead of 'people.

Answer: The term has been substituted.

L141-146. This paragraph is confusing, explain clearly that it refers to coprological analysis and specify how many times each individual was tested.

Answer: This sentence has been added: Several coprological analyzes were performed on each individual throughout the period of stay in the WRC.

L164-165: “One-way analysis of variance and non-parametrical Mann-Whitney U test were used to compare means” explain in which situations the Mann-Whitney test was used instead of ANOVA.

Answer: This a general sentence: if the variables have a normal distribution, we use the one-way analysis of variance; if not, the non-parametrical M-W. If necessary, we will specify wich test we have used in the results section.

L167-168: “A logistic regression model (LRM) was carried out to determine which predictor 167 variables were associated with unassisted mortality” It should be clarified which variables are considered predictor variables. Based on Table 3, it may be inferred that the predictor variables include the presence of a clinical sign (as the odds ratio is presented in this table). However, it is unclear whether the predictor variables also include those listed in Table 6 or a combination of both. This ambiguity is confusing and should be explicitly stated in the description of the statistical methods.

Answer: The predictors included in the multivariable LRM were showed in table 6. The Statistic section have been modified according to your comment as follows:

“Predictor variables included at the LRM were codified as binary variables: sex, age, fecal parasites, and clinical signs grouped as: general signs, integument, neurological and musculoesketal. Furthermore, two categorical variables were included: body condition (obesity, normal, thin, emaciation) and PCV (low, normal, high).”

L172: “the independent variables” They have been called previously predictor variables, keep uniformity in terms used.

Answer: The term has been substituted.

L173-174: What threshold of the Multiple Correlation Coefficient was used to determine the presence of collinearity?

Answer: We have deleted this sentence. We normally consider a value of 0.6.

L174: “multivariate analyses”. There is not clear to which analysis the authors refer. There are lots of types of multivariate analyses. Presumably the authors mean a Logistic regression with several predictors. This should be stated clearly.

Answer: The type of multivariate analyses has been specified: “before the application of the multivariate LRM”

L199-202: “had statistical 199 significant higher frequencies in Algerian hedgehog compared to European hedgehog 200 (χ2=21.5 p=0.001)”. “Moreover, males were a(t) major risk of “Trauma” and “Natural disease” 201 than females in both species (χ2=11.5; p=0.021)”. The authors base their affirmations relative to differences in specific causes of admission on the X2 test. However, these tests only inform that the distributions of causes differ between species or sexes, respectively, not on the particular causes that generate the difference. In fact, the authors have based their statements on differences in specific causes on the comparison of the confidence intervals of the percentage of each cause. Therefore, the text requires correction and improvement.

Answer: In Algerian hedgehog the frequency of “Orphaned/young” (50.3%) and “Other causes” (15.6%) were statistical significant higher (χ2=21.5; df=4; p<0.001). On the other hand, “Trauma” and “Natural disease” were more frequent in males in both species (χ2=11.5; df=4; p=0.021).

Furthermore, there is a lack of explanation regarding how the confidence intervals were calculated.

Answer: Line 163: CI were calculated using the Macro “Confidence Interval for Proportions !CIP V2003.07.15 (c) A.Bonillo, JM.Domenech & R.Granero”.

L220: Mention of “Statistically significant differences (p<0.05). Degree of freedom (df)= 4.” does not make sense without specifying the type of statistical test to which it refers.

Answer: Chi-square test (χ2) with 4 Degree of freedom (df).

L235: When you mention 632 cases, 321, etc., are you referring to individuals? If so, make it clear that 632 individuals had a single fecal exam, and so on.

Answer: The term has been substituted.

L245-247: The degrees of freedom for the X2 test should be provided. Additionally, maintain consistency in the number of decimals for p-values throughout the paper. If a p-value is very low, it may be presented as p<0.001.

Answer: (df) added.df=1.

Numbers of decimals corrected: 3 decimals.

Table 4. header “Prevalence (IC 95%)”. IC is in Spanish, it should be CI.

Answer: The term has been corrected.

Table 4: This table is somewhat confusing and requires a clearer explanation. Do 'First,' 'Second,' etc., tests refer to the first test, second test, etc.? If so, why do you mention a maximum of 4 tests in line 236, while the table indicates 5? Adding sample sizes in the header of each column would enhance clarity. Additionally, the rationale for presenting this table is not evident, as the information it provides is not used in a clear manner. Furthermore, the paragraph from lines 243-248 makes comparisons based on age, but age is not shown in Table 4.

Answer: I totally agree with you. The table was confusing and the rationales was not defined clearly. In fact, we have decided to present only the overall prevalence. That means counting only the first positive result in any of the fecal analyzes. Further and deeper studies are necessary to compare the changes of prevalence among the stay at the center or the response to antiparasitic therapy, and this can be the object of another paper.

We have corrected the number of fecal analyzes per individual, including individuals with a 5th coprological test. I miss this column in the original excel file. Thank you for your comment, again.

Parasitic infection, sex and age were codified as binary variables. We used  2x2 tables and the Chi Square for comparison of proportions. In example, the SPSS syntax for parasitological was: “CROSSTABS /TABLES= SEXDET ANYCALDET BY COPRO1BIN CAP1 TREM1 COC1 COPROSFINALALTRES LP1 R1 INF_MULT1 COPRO2BIN CAP2 TREM2 COC2 ALTRES2 LP2 COPROSFINALR2 INF_MULT2 R3 INF_MULT3 R4 INF_MULT4 R5 INF_MULT5 RES1 RES2 RES3 RES4 RES5 PARASITATS.

Finally, we have only presented the results statistically significant in the text in order to simplify the results section.

L266-270 and table 5: Statistical analysis compare adult and young individuals but age is not included in table 5. X2 tests do not include degrees of freedom. In addition, comments on X2 test in L199-202 are also applicable here.

Answer: We have included the (df).

Macroscopic lesions, sex and age were codified as binary variables. We used 2x2 crosstabs and we decided to present in the text (not in a table or in figure 3) the statistical significant results. (In fact, you can read the same comment above).

L271-273: no test for such sentence, only some unexplained asterisks in figure 3.

Answer: This was a mistake. The asterisks were removed from the figure.

L280: “In the bivariate model, females and tested negative for parasites were protective factors.” Sentence difficult to understand, rewrite.

Answer: We have changed the sentence: “In the bivariate model, all variables were associated with a worse prognosis, except being female or not being parasitized.”

Table 6. L288-290: this is confusing, I assume that refers to the reference level of each factor, but this should be clarified.

Answer: Effectively, it refers to the reference level of each factor. In the statistical section we have shortly explained how we have codified the predictors, making the understanding of Table 6 clearer.

Round 3

Reviewer 3 Report

Comments and Suggestions for Authors

The authors have addressed most of the recommendations I made and I think that the article may be now accepted. However, there are some minor issues, so I recommend the following modifications

L169-179: change redaction to “One-way analysis of variance or the non-parametric Mann-Whitney U test, depending on the distribution of each variable (whether it is normal or not), was used to compare means”

L179-: “Predictor variables included at the LRM 179 were codified as binary variables: sex, age, presence of fecal parasites and clinical …” add presence of

L261: For clarity, it would be better to include the sample size used to calculate prevalence in the table 4.

L302-304. Table 6. The explanation of variables remains unclear. I recommend either removing them in their current form or rewriting for greater clarity

Comments on the Quality of English Language

OK

Author Response

Dear reviewer: thank you very much for your comments. Without them, the current version does not be possible. It help us a lot.

As well, results are summarized in the text: “A logistic regression analysis was performed to estimate the survival of the animals admitted for rehabilitation. Descriptive parameters of the prognostic factors are presented in Table 6. In the bivariate model, all variables were associated with a worse prognosis, except being female or not being parasitized.”

The new revised version the footnote of Table 6 now appears as: “ns, no significant differences (p>0.05). § Multivariate regression”.